# OpenReview forum: "Diff-BBO:  Diffusion-Based Inverse Modeling for Black-Box Optimization"
_ICLR.cc/2025/Conference — Submitted to ICLR 2025_

### Official Review · Reviewer_TAkd · 2024-10-29

**Soundness:** 2
**Presentation:** 2
**Contribution:** 2
**Rating:** 6
**Confidence:** 2

**Summary:**

In this submission, the authors tackle the challenge of black-box optimization. At present, the bulk of the field has concentrated on methods like Bayesian Optimization, which employ a _forward_ model $p(y|\mathbf{x})$ and an acquisition function $\alpha(\mathbf{x})$. Maximizing the latter at each iteration $t$ yields a candidate design $\mathbf{x}_t$ that will in turn give a noisy evaluation $y_t$.

Interestingly, here, the authors use an _inverse_ model $p(\mathbf{x}|y)$ together with an acquisition function $\tilde{\alpha}(y)$. As such, one can learn to generate designs $\mathbf{x}$ conditionally to an output value $y$, such that the evaluated design results in a noisy evaluation $y_t \approx y$. The precise value of $y$ that should be used for conditional generation is obtained by maximizing the acquisition function $\tilde{\alpha}$.

In this work, conditional generation is achieved through conditional diffusion models. While diffusion models have been used in black box optimization previously, building a suitable acquisition function to handle these models has not been done yet. The latter must adequately trade-off exploitation (high values $y$) and exploitation (values $y$ for which uncertainty model uncertainty $p(\mathbf{x}|y)$ is high). The paper's main contribution is a thorough study of uncertainty quantification for conditional diffusion models, leading to a decomposition between _aleatoric_ and _epistemic_ uncertainty. This ultimately leads to an acquisition function that trades off high function values and epistemic uncertainty.

Finally, the performance of the proposed acquisition function is theoretically grounded depending on some assumptions on the target function $f$, and the method itself is shown to outperform concurrent baselines on a number of continuous and discrete datasets.

**Strengths:**

- I found the paper to be well-organized and motivated. The technical novelty is light but seems to be theoretically grounded.

- The proposed method consistently ranks among the best competitors on a benchmark involving multiple baselines and datasets, while maintaining computation times in the same order of magnitude as Gaussian Process-based alternatives.

**Weaknesses:**

- From a practical point of view, I am not sure that Theorems 2 and 3 are useful: they assume the $L$-smoothness of the function $f$. This function takes as input $\x$, which might be discrete, or an embedding of a discrete input like a molecule, and there is little to no chance to have $L$-smoothness in this case unless the embedding explicitly enforces that assumption. I believe this should at least be mentioned.

**Questions:**

- An interesting ablation study would be to add a factor weight $\beta$ in front of the epistemic uncertainty in Equation 8, and to vary this term. This would give an insight into how important uncertainty is in the acquisition process.

---

> ### Author Response · Authors · 2024-11-21
> **Response to Reviewer TAkd (1/2)**
>
> We thank the reviewer for the constructive comments and thoughtful suggestions. We address the concerns as follows:
>
> > From a practical point of view, I am not sure that Theorems 2 and 3 are useful: they assume the 𝐿-smoothness of the function 𝑓. This function takes as input \x, which might be discrete, or an embedding of a discrete input like a molecule, and there is little to no chance to have 𝐿-smoothness in this case unless the embedding explicitly enforces that assumption.
>
> **We would like to clarify that our work does not assume L-smoothness; instead, we assume function $f$ satisfies Lipschitz continuity**, which is a weaker and more realistic condition in practical settings. The definition of Lipschitz continuity, as provided in Appendix B, states that
>     \begin{equation}
>         \left| f(\boldsymbol{x}^\prime) - f(\boldsymbol{x})\right| \leq L \Vert\boldsymbol{x}^\prime - \boldsymbol{x}\Vert, ~~\forall \boldsymbol{x}^\prime, \boldsymbol{x} \in \mathbb{R}^d.
>     \end{equation}
> Unlike L-smoothness, which involves a second-order differentiability condition, Lipschitz continuity only relates to the rate of change of $f$ and does not impose additional smoothness constraints. Under this definition, the input need not to be continuous (i.e. input can also be discrete). **Lipschitz continuity is a mild and realistic assumption that is commonly satisfied in practical scenarios, even with deep neural networks** [1][2]. Our empirical results on diverse tasks—including discrete ones like TFBind8 and TFBind10— also demonstrate that these theoretical guarantees translate into robust and effective optimization performance in practice.
>
> This assumption is imposed for theoretical analyses. We do not assume a perfect diffusion model, which necessitates to bound the difference in the objective space ($f(\boldsymbol{x})$) with that in the design space ($\boldsymbol{x}$), so as to establish meaningful guarantees for the sub-optimality performance gap. It helps quantify how changes in the design space (given by samples generated from diffusion models) affect the objective values.
>
> Furthermore, we would like to emphasize the takeaways of our theorems to form a better understanding for practitioners:
> * Quantifying sub-optimality: Theorem 2 establishes that the expected sub-optimality gap is bounded. This means that Diff-BBO provides a principled way to ensure that the generated candidates are close to optimal in expectation.
> * Controlling variance through uncertainty: Theorem 3 highlights that the variance of the sub-optimality gap is influenced by both aleatoric (inherent noise) and epistemic (model uncertainty) components. This decomposition underscores the importance of reducing epistemic uncertainty to stabilize the optimization outcomes and enhance reliability.
> * Balancing exploration and exploitation: Theorem 4 provides a theoretical foundation for the design of our acquisition function, Uncertainty-aware Exploration (UaE), which balances targeting high objective values and minimizing epistemic uncertainty. This balance enables practitioners to efficiently explore the design space while ensuring reliable convergence to optimal solutions.
>
> [1] Bartlett, Peter L., Dylan J. Foster, and Matus J. Telgarsky. "Spectrally-normalized margin bounds for neural networks." Advances in neural information processing systems 30 (2017).
>
> [2] Fazlyab, Mahyar, et al. "Efficient and accurate estimation of lipschitz constants for deep neural networks." Advances in neural information processing systems 32 (2019).

---

> ### Author Response · Authors · 2024-11-21
> **Response to Reviewer TAkd (2/2)**
>
> > An interesting ablation study would be to add a factor weight 𝛽 in front of the epistemic uncertainty in Equation 8, and to vary this term. This would give an insight into how important uncertainty is in the acquisition process.
>
> Good suggestion. We have conducted additional ablation studies on the factor weight $\beta$ for both discrete task (TFBind8) and continuous task (Ant), with $\beta \in \\{0.6, 0.8. 1.0, 1.2, 1.4\\}$. The results are averaged across three random runs, presented in Table A and Table B.  The results show that when $\beta$ value is too low (e.g., $\beta=0.6$), the acquisition function struggles to dynamically determine which $y$ to condition on and keep selecting the highest $y$ with high likelihood. This fixed condition approach results in suboptimal solutions, aligning with our findings from the fixed-condition ablation study. Fine-tuning $\beta$ can also enhance performance; for instance, results in Table B (Ant) show that $\beta = 1.2$ outperforms $\beta = 1.0$.
>
> Table A: TFBind8
> | iteration | 1         | 3         | 5         | 7         | 9         | 11       | 13        | 15        |
> |-----------|-----------|-----------|-----------|-----------|-----------|----------|-----------|-----------|
> | $\beta$ = 0.6       | 0.868     | 0.891     | 0.891     | 0.891     | 0.904     | 0.956    | 0.968     | 0.971     |
> | $\beta$ = 0.8       | **0.886** | 0.905     | 0.924     | 0.942     | 0.965     | 0.965    | 0.971     | 0.978     |
> | $\beta$ = 1         | 0.869     | 0.892     | **0.969** | **0.969** | **0.969** | **0.97** | **0.972** | **0.979** |
> | $\beta$ = 1.2       | 0.868     | 0.905     | 0.946     | 0.965     | 0.967     | 0.967    | 0.967     | 0.976     |
> | $\beta$ = 1.4       | 0.855     | **0.933** | 0.936     | 0.936     | 0.952     | 0.967    | 0.97      | 0.974     |
>
>
>
> Table B: Ant
> | iteration | 1          | 3          | 5          | 7          | 9          | 11         | 13         | 15         |
> |-----------|------------|------------|------------|------------|------------|------------|------------|------------|
> | $\beta$ = 0.6       | 291.69     | 354.85     | **391.72** | 391.72     | 420.43     | 435.39     | 442.8      | 460.66     |
> | $\beta$ = 0.8       | 346.98     | **375.36** | 380.49     | 439.22     | 439.22     | 463.52     | 463.52     | 468.13     |
> | $\beta$ = 1         | **368.99** | 368.99     | 376.23     | 395.16     | 403.58     | 459.03     | 470.22     | 470.22     |
> | $\beta$ = 1.2       | 339.69     | 374.9      | 374.9      | **441.95** | **466.63** | **472.82** | **483.53** | **493.08** |
> | $\beta$ = 1.4       | 297.89     | 350.19     | 366.05     | 394.77     | 421.02     | 463.16     | 463.16     | 473.02     |

---

> > ### Author Response · Authors · 2024-11-27
> > **The end of the rebuttal phase is approaching**
> >
> > Dear Reviewer TAkd,
> >
> > As the end of the rebuttal phase is approaching, we would like to kindly confirm if our responses have successfully addressed your concerns and clarified your questions. If you have any additional questions or concerns, we would be happy to address them.
> >
> > Best regards,
> >
> > Authors of Diff-BBO

---

> > > ### Comment · Reviewer_TAkd · 2024-11-27
> > > **Answer received!**
> > >
> > > I have read the authors' rebuttal as well as their answers to all reviewer's questions, including mine. I am keeping my score as is.

---

### Official Review · Reviewer_cYdB · 2024-10-29

**Soundness:** 2
**Presentation:** 4
**Contribution:** 3
**Rating:** 5
**Confidence:** 3

**Summary:**

This paper proposes a novel approach for solving black-box optimization problems. Unlike traditional methods that focus on learning a surrogate to evaluate design decision quality, this approach employs a diffusion model to approximate the distribution within the design space conditioned on a target value. Unlike existing inverse methods that assume access to an offline dataset, this paper studies a dynamic setting where the exploration-exploitation tradeoff must be considered. To address this challenge, an uncertainty-aware exploration method is introduced. The effectiveness of the proposed approach is demonstrated through extensive numerical studies.

**Strengths:**

- Overall the paper was very well-written. The key concepts and challenges are clearly introduced, which makes it easy for me to appreciate the contribution.

- Black-box optimization is an important methodology that has a wide range of applications in engineering and science.

- The numerical studies are comprehensive and convincing.

**Weaknesses:**

- **Soundness of the theoretical analysis**. While I appreciate the authors efforts to justify the proposed approach through a theoretical lens, I found some of the results unsatisfying.

  - For example, in Theorem 2, the bound does not depend on the number of samples collected in each iteration $N$. Intuitively, a large $N$ might lead to over-conservative estimates and a small $N$ might renders the estimate too optimistic. Relating the this bound to $N$ may lead to insights into the choice of this important hyper-parameter. The current bound is independent of $N$, suggesting that it might be loose. Furthermore, this theorem assumes the existence of a perfect diffusion model. I suggest the authors add further discussion on the implication/validity of this assumption. This comment applies to Theorem 3 as well.
  - Under a similar Lipschitz assumption, is it possible to derive surrogate approximation guarantees for the forward-based approach? If so, how does the inverse bound compare to the forward bounds?


- **Acquisition function.** This is a minor point, but I was wondering if a weight should be assigned to  $\Delta$ in Equation (8) because (1) the two terms might be in different scales, and (2) the users may dynamically adjust the weight across different iterations to balance exploration and exploitation.

**Questions:**

See weaknesses.

---

> ### Author Response · Authors · 2024-11-21
> **Response to Reviewer cYdB (1/3)**
>
> We thank the reviewer for the constructive comments and thoughtful suggestions. We address the concerns as follows:
>
> > in Theorem 2, the bound does not depend on the number of samples collected in each iteration 𝑁. Intuitively, a large 𝑁 might lead to over-conservative estimates and a small 𝑁 might renders the estimate too optimistic. Relating the this bound to 𝑁 may lead to insights into the choice of this important hyper-parameter. The current bound is independent of 𝑁, suggesting that it might be loose.
>
> **We did consider $N$ and assigned $N = 1$, as noted in Appendix B (line 861).** We provide the clarification on the role of $N$ in our analysis and its implications for Theorem 4 below.
> * Purpose of Theorem 2 and 3: Theorem 2 and 3 are designed to provide **a general bound** on the sub-optimality performance gap that **holds under all circumstances**, independent of specific hyperparameter choices such as $N$. **This generality is necessary to support Theorem 4**, which justifies the validity of our acquisition function, Uncertainty-aware Exploration (UaE). By providing a bound that does not depend on $N$, we ensure that Theorem 4 remains robust and valid for any $N$, including cases where only a single sample ($N = 1$) is used in each iteration.
> * In our proofs, we assume $N = 1$ to simplify the analysis. This assumption aligns with the theoretical focus of Theorem 2 and 3 on providing a general bound.
> * Problems with high-probability bounds: Extending the analysis to $N > 1$ would require incorporating high-probability bounds, which account for the variability introduced by multiple samples. While such bounds can provide additional insights, they do not always hold universally due to their dependence on specific statistical properties of the sampling process (e.g., tail bounds, variance). As a result, further assumptions will be required in order to have a tighter bound. However, as we explained earlier, this does not satisfy our purpose of supporting the claims of Theorem 4. Besides, when $N$ is small, the estimates may be overly optimistic due to insufficient exploration, whereas a large $N$ might lead to conservative estimates that dilute the benefits of targeted sampling. These trade-offs would require further assumptions and could complicate the generality of Theorem 4.
> * Insights into practical hyperparameter choices: Although Theorem 2 provides a bound independent of $N$, its practical implications remain valid for guiding the design of the acquisition function.
>
> > this theorem assumes the existence of a perfect diffusion model. I suggest the authors add further discussion on the implication/validity of this assumption.
>
> There may be a misunderstanding. Our theoretical analysis does not assume the existence of a perfect diffusion model. That’s why we need to introduce the Lipschitz continuity assumption to consider the sampling errors brought by the diffusion model. It ensures that the sub-optimality performance gap can be meaningfully bounded, even when the diffusion model does not perfectly reconstruct the target conditional values $y_k^*$ in each iteration $k$.  In particular, we bound the error in the value space ($f(\boldsymbol{x})$) with that in the design space ($\boldsymbol{x}$), so as to establish meaningful guarantees for the sub-optimality performance gap and its variance. It helps quantify how changes in the input design space (given by samples generated from diffusion models) affect the objective values.

---

> > ### Comment · Reviewer_cYdB · 2024-11-23
> >
> > Thank you for the clarification. I've read the response. However, I still have concerns about the bounds derived in this paper. Simply setting $N=1$ is different from considering $N$. I understand the difficulty of doing theoretical analysis when considering a general $N$. However, in its current form, the so-called "universal" bounds offer limited insights. We only know there "exists" a bound, yet we have limited information about how tight the bound is, or if there is anything we can learn from the bound. For these reasons, I decided to keep my original rating.

---

> > > ### Author Response · Authors · 2024-11-24
> > > **Response to Reviewer cYdB**
> > >
> > > We thank the reviewer for the response. We want to clarify that $N=1$ represents the loosest bound scenario. Therefore, the bound provided in Theorem 2 serves as a general bound on the sub-optimality performance gap that holds under all circumstances. We would like to emphasize again that the purpose of providing this general bound is to support the designed acquisition function, Uncertainty-aware Exploration (UaE). **Theorems 2 and 3 are provided to support Theorem 4, where we proved that Diff-BBO with UaE achieves a near-optimal solution for the online BBO problem even in the case when $N=1$. This is the key insight we can learn from the bound.** It is unreasonable to judge the soundness of Theorems 2 and 3 solely based on the tightness of the bound, without considering their purpose and their relationship with Theorem 4.

---

> ### Author Response · Authors · 2024-11-21
> **Response to Reviewer cYdB (2/3)**
>
> > Under a similar Lipschitz assumption, is it possible to derive surrogate approximation guarantees for the forward-based approach? If so, how does the inverse bound compare to the forward bounds?
>
> We thank the reviewer for raising this question. While this is an intriguing direction, we would like to clarify the following points:
>
> * Incomparable purposes: Forward approaches and our inverse modeling framework serve **fundamentally different purposes**, making a direct comparison of bounds less meaningful. Forward methods typically aim to model the unknown objective function $f$, while our approach focuses on inverse modeling to propose optimal candidates. Consequently, the theoretical results derived for our inverse approach are tailored to justify the design and validity of our acquisition function, Uncertainty-aware Exploration (UaE), rather than to compete with or replicate bounds from forward methods.
> * Heuristic nature of forward acquisition functions: **Most acquisition functions in forward approaches are based on heuristics**, rather than bounds grounded in surrogate approximation guarantees. While forward-based methods are effective in practice, their theoretical guarantees are typically unavailable. The most aligned results are only available in bandit literature rather than the online black-box optimization literature, rendering highly-different problem domains. While deriving theoretical guarantees for forward-based approaches under similar assumptions could be interesting, such an exploration is far beyond the scope of this paper.
> * **The primary goal of our theoretical analysis is not to derive state-of-the-art tight bounds but to provide a sound justification for the design of UaE.** Our results establish that UaE balances exploration and exploitation effectively, leveraging both the inherent properties of conditional diffusion models and the epistemic uncertainty to achieve robust optimization performance.
>
> Furthermore, we would like to emphasize the takeaways of our theorems to form a better understanding for practitioners:
>
>
> * Quantifying sub-optimality: Theorem 2 establishes that the expected sub-optimality gap is bounded. This means that Diff-BBO provides a principled way to ensure that the generated candidates are close to optimal in expectation.
> * Controlling variance through uncertainty: Theorem 3 highlights that the variance of the sub-optimality gap is influenced by both aleatoric (inherent noise) and epistemic (model uncertainty) components. This decomposition underscores the importance of reducing epistemic uncertainty to stabilize the optimization outcomes and enhance reliability.
> * Balancing exploration and exploitation: Theorem 4 provides a theoretical foundation for the design of our acquisition function, Uncertainty-aware Exploration (UaE), which balances targeting high objective values and minimizing epistemic uncertainty. This balance enables practitioners to efficiently explore the design space while ensuring reliable convergence to optimal solutions.

---

> ### Author Response · Authors · 2024-11-21
> **Response to Reviewer cYdB (3/3)**
>
> > This is a minor point, but I was wondering if a weight should be assigned to Δ in Equation (8) because (1) the two terms might be in different scales.
>
> Good suggestion. We added a factor weight $\beta$ in front of the epistemic uncertainty in Equation 8 and conducted additional ablation studies for both discrete task (TFBind8) and continuous task (Ant), with $\beta \in \\{0.6, 0.8. 1.0, 1.2, 1.4\\}$. The results are averaged across three random runs, presented in Table A and Table B.  The results show that when $\beta$ value is too low (e.g., $\beta=0.6$), the acquisition function struggles to dynamically determine which $y$ to condition on and keep selecting the highest $y$ with high likelihood. This fixed condition approach results in suboptimal solutions, aligning with our findings from the fixed-condition ablation study. Fine-tuning $\beta$ can also enhance performance; for instance, results in Table B (Ant) show that $\beta = 1.2$ outperforms $\beta = 1.0$.
>
> Table A: TFBind8
>
> | iteration | 1         | 3         | 5         | 7         | 9         | 11       | 13        | 15        |
> |-----------|-----------|-----------|-----------|-----------|-----------|----------|-----------|-----------|
> | $\beta$ = 0.6       | 0.868     | 0.891     | 0.891     | 0.891     | 0.904     | 0.956    | 0.968     | 0.971     |
> | $\beta$ = 0.8       | **0.886** | 0.905     | 0.924     | 0.942     | 0.965     | 0.965    | 0.971     | 0.978     |
> | $\beta$ = 1         | 0.869     | 0.892     | **0.969** | **0.969** | **0.969** | **0.97** | **0.972** | **0.979** |
> | $\beta$ = 1.2       | 0.868     | 0.905     | 0.946     | 0.965     | 0.967     | 0.967    | 0.967     | 0.976     |
> | $\beta$ = 1.4       | 0.855     | **0.933** | 0.936     | 0.936     | 0.952     | 0.967    | 0.97      | 0.974     |
>
> Table B: Ant
> | iteration | 1          | 3          | 5          | 7          | 9          | 11         | 13         | 15         |
> |-----------|------------|------------|------------|------------|------------|------------|------------|------------|
> | $\beta$ = 0.6       | 291.69     | 354.85     | **391.72** | 391.72     | 420.43     | 435.39     | 442.8      | 460.66     |
> | $\beta$ = 0.8       | 346.98     | **375.36** | 380.49     | 439.22     | 439.22     | 463.52     | 463.52     | 468.13     |
> | $\beta$ = 1         | **368.99** | 368.99     | 376.23     | 395.16     | 403.58     | 459.03     | 470.22     | 470.22     |
> | $\beta$ = 1.2       | 339.69     | 374.9      | 374.9      | **441.95** | **466.63** | **472.82** | **483.53** | **493.08** |
> | $\beta$ = 1.4       | 297.89     | 350.19     | 366.05     | 394.77     | 421.02     | 463.16     | 463.16     | 473.02     |

---

### Official Review · Reviewer_DvmK · 2024-11-03

**Soundness:** 3
**Presentation:** 3
**Contribution:** 3
**Rating:** 5
**Confidence:** 2

**Summary:**

In traditional BBO a surrogate model $\hat{f}$ is learned to approximate the objective function and then helps optimize an acquisition function which yields the next $x_{k+1}$ where to evaluate $f$. To select the next query point $x_{k+1}$ Diff-BBO instead performs posterior inference in the parameter space $X$ conditioned on a specified target objective value $y$ obtained by maximising an introduced Uncertainty-aware Exploration (UaE) acquisition function.

Conditional diffusion model are trained to learn the conditional distribution $p(x | y)$, where $x$ represents feasible inputs in the parameter space.

$y$ is chosen with a proposed acquisition function, Uncertainty-aware Exploration (UaE), that prioritizes target values $y$ with high expected objective values while minimizing epistemic uncertainty. This acquisition function balances exploration and exploitation. The paper provides theoretical proofs demonstrating that UaE achieves a near-optimal solution for the BBO problem.

Numerical experiments to support the work are presented

**Strengths:**

The paper introduces a new approach supported by solid theoretical results and empirical validation across diverse tasks. Difference with existing methods is clearly presented.

**Weaknesses:**

An important part of the procedure is how is assembled the candidate set $\mathcal{Y}$ and its corresponding weights $w$. The paper does not specify a principled method for choosing or tuning these weights, which makes this important aspect somewhat empirical given that for too high weights, the model may focus excessively on unfeasibly high values of $y$ while too low weights might limit the search space.


A diffusion model requires a large dataset to effectively learn the data manifold in the design space. If the function $f$ is expensive to evaluate, building a large dataset may be computationally expensive. Most of the experiments are run with a relatively high number of evaluations. How would the method perform on a smaller dataset?

Diffusion models are usually susceptible to mode collapse, where generated samples fail to cover the full distribution of the data. Was this observed? This could cause Diff-BBO to overlook potentially optimal regions in the design space.

**Questions:**

Is there a systematic approach for choosing the weights $w$ for the candidate set $\mathcal{Y}$, or is this step largely empirical?

Could the method perform effectively with a smaller accumulated dataset, as opposed to the relatively high number of evaluations used in the experiments?

Was this issue of mode collapse observed in Diff-BBO? If so, how does it impact the model’s ability to explore potentially optimal regions in the design space?

---

> ### Author Response · Authors · 2024-11-21
> **Response to Reviewer DvmK**
>
> We thank the reviewer for the constructive comments and thoughtful suggestions. We address the concerns as follows:
>
> > An important part of the procedure is how is assembled the candidate set 𝑌 and its corresponding weights 𝑤. The paper does not specify a principled method for choosing or tuning these weights, which makes this important aspect somewhat empirical given that for too high weights, the model may focus excessively on unfeasibly high values of 𝑦 while too low weights might limit the search space.
>
> > Is there a systematic approach for choosing the weights 𝑤 for the candidate set 𝑌, or is this step largely empirical?
>
> We would like to clarify that **selecting the weights $w$ is equivalent to select the objective values $y$** since $y = w.\phi_k$ (line 240), where $\phi_k$ is a predetermined constant ($\phi_k$ is the maximum function value being queried in the current training dataset D (line 241-242)). We did provide a principled method for choosing $y$ (equivalently choosing weight $w$) using the acquisition function uncertainty-aware exploration (UaE). Specifically, we identify the optimal scalar value $y^*$ as $y^*= \text{argmax}_y \alpha(y,D)$, where $\alpha(y,D)= y- \text{epistemic}(y, D)$ (Eqn 8). We employ this acquisition function to ensure safe exploitation on high $y$ values. Additionally, Diff-BBO leverages the advantages of the diffusion model, utilizing the stochastic nature of the diffusion process to explore the design space. Therefore, low values of $y$ or $w$ do not limit the search space.
>
> > A diffusion model requires a large dataset to effectively learn the data manifold in the design space. If the function 𝑓 is expensive to evaluate, building a large dataset may be computationally expensive. Most of the experiments are run with a relatively high number of evaluations. How would the method perform on a smaller dataset?
>
> > Could the method perform effectively with a smaller accumulated dataset, as opposed to the relatively high number of evaluations used in the experiments?
>
> We agree that training a deep generative model such as diffusion model requires a relatively large dataset in learning the data distribution. However, this data requirement is only required at the initial stage. Meanwhile Diff-BBO does not demand high-quality data for the initial training set. In our experimental setup, we only include the data with below-average objective scores from the offline dataset of design-bench in the initial training set (line 427-429). This mirrors a common scenario in real-world experimental design tasks, where preexisting and unoptimized data is available in databases. At the active learning stage, The experiment results shows that Diff-BBO is much more sample-efficient compared with other baselines (shown on Figure 3). Furthermore, we conducted an ablation study (shown on Figure 5) to verify that Diff-BBO is robust w.r.t the varying batch sizes, ranging from 25 to 200.
>
> > Diffusion models are usually susceptible to mode collapse, where generated samples fail to cover the full distribution of the data. Was this observed? This could cause Diff-BBO to overlook potentially optimal regions in the design space.
>
> > Was this issue of mode collapse observed in Diff-BBO? If so, how does it impact the model’s ability to explore potentially optimal regions in the design space?
>
> We did not encounter the mode collapse issue when using the conditional diffusion model. It is also important to note that prior studies indicate that **diffusion models are more robust against mode collapse compared to other generative models, such as GANs** [1][2][3]. The stochastic nature of the diffusion process enables the model to cover the data distribution more comprehensively. As a result, it can capture the diversity of the dataset and generate versatile samples. This capability is particularly beneficial for our Diff-BBO algorithms to enhance the exploration of the design space.
>
> [1] Ulhaq, Anwaar, and Naveed Akhtar. "Efficient diffusion models for vision: A survey." arXiv preprint arXiv:2210.09292 (2022)
>
> [2] Croitoru, Florinel-Alin, et al. "Diffusion models in vision: A survey." IEEE Transactions on Pattern Analysis and Machine Intelligence 45.9 (2023): 10850-10869.
>
> [3] Kazerouni, Amirhossein, et al. "Diffusion models in medical imaging: A comprehensive survey." Medical Image Analysis 88 (2023): 102846.

---

> > ### Comment · Reviewer_DvmK · 2024-11-25
> >
> > Thank you for your response.
> >
> > As it stands, most of my concerns were not answered.
> > In the same order:
> > - My question was about building the candidate set $Y$. Once $\phi_k$ is selected, how is the set $Y$ constructed? It seemed to me a set of weights $w$ was used in order to construct $Y$. If not, could the authors clarify this points?
> > - I was aware of the ablation study shown on Figure 5. My questions still stands. I don't see how in the case where few data samples are gathered at each step it is possible to adequately update a diffusion model to incorporate any information. I suspect all the information is already present at what the authors call the "initial stage" in the initial dataset.
> >
> > I have decided to keep my score

---

> > > ### Author Response · Authors · 2024-11-26
> > > **Response to Reviewer DvmK**
> > >
> > > We thank the reviewer for the response. We address the concerns as follows:
> > >
> > > > My question was about building the candidate set $Y$. Once $\phi_k$ is selected, how is the set $Y$ constructed? It seemed to me a set of weights $w$ was used in order to construct $Y$. If not, could the authors clarify this points?
> > >
> > > Constructing the candidate set $Y$ is identical to forming the weight set $W$. Sampling from the one-dimensional objective space is much more convenient than sampling from the high-dimensional design space. In practice, we utilize the weight set $W =\\{0.6, 0.7, 0.8, 0.9, 1.0\\}$. We employ a principled method to select the weight $w$ using the uncertainty-aware exploration (UaE) acquisition function. Additionally, our ablation study depicted in Figure 4 indicates that conditioning on a large $w$, significantly greater than 1, does not enhance the performance. Therefore, it is sufficient to sample $w$ around 1.
> > >
> > > > I don't see how in the case where few data samples are gathered at each step it is possible to adequately update a diffusion model to incorporate any information. I suspect all the information is already present at what the authors call the "initial stage" in the initial dataset.
> > >
> > > Whether it is adequate to update the diffusion model do depend on the batch size. For instance, retraining the model at each iteration is unnecessary when the batch size is 1 and the size of initial dataset is relatively large. This prompted us to conduct the ablation study on batch sizes. As shown in Figure 5, with batch sizes of 25 and 50, we observed several iterations where the maximum objective score remained unchanged. However, when the batch size was increased to 100, these issues were largely resolved. It is clear that the conditional diffusion model has been updated from the newly generated dataset, as indicated by the continuous increase in the maximum objective value. This trend is consistent across all tasks depicted in Figure 3.
> > >
> > > The significant performance improvement from the initial to the final iteration demonstrates that the conditional diffusion model does benefit from updates with the added dataset. Although the overall number of samples may appear large compared to low-dimensional Bayesian Optimization tasks, this is a necessary requirement for addressing high-dimensional scientific design problems. And we ensured a fair comparison in our experiment by keeping the total number of samples consistent across all baselines and our approach.
> > >
> > > Please let us know if there are any other follow-up questions.

---

### Official Review · Reviewer_1x7Q · 2024-11-04

**Soundness:** 2
**Presentation:** 2
**Contribution:** 2
**Rating:** 3
**Confidence:** 5

**Summary:**

This work proposes an inverse approach leveraging diffusion models for online BBO problem. Specifically, this paper introduces a new acquisition function to propose objective function values, and employ a conditional diffusion model to generate samples. The authors conduct experiments in design-bench to verify the effectiveness of their method.

**Strengths:**

1. This paper is easy to follow. The structure of this paper is clear.
2. This paper provides a solution for online black-box optimization (BBO) called Diffusion-based inverse modeling for black-box optimization (DIFF-BBO), which uses objective function values to generate solutions. This approach is interesting and has the potential to improve the performance of online BBO.

**Weaknesses:**

1. The core difference between the main idea of the proposed method (the inverse method) and LLAMBO [1] should be explicitly discussed.  While there are differences in the specific implementation details compared to LLAMBO, this approach appears to be more of an aggregation of the previous methods. Specifically, the inverse approach was originally utilized in MINS [2], while the conditional diffusion model was incorporated in DDOM [3].

2. Why do the experimental tasks of online BBO methods use offline BBO benchmarks? Is it because there is no suitable benchmark for online BBO?

3. While the background on online BBO is quite solid, the paper does not sufficiently explore prior work in offline BBO. Expanding the discussion to include them could provide a more comprehensive literature review and motivation support.

4. Superconductor, Ant and D’Kitty and so on are high dimensional problem. So, it would be better if this paper could compare with more high dimensional Bayesian optimization in recent years rather than simple black-box optimization.

[1] Large Language Models to Enhance Bayesian Optimization. In Proceeding of the 12th International Conference on Learning Representations, Vienna, Austria, 2024.

[2] Model inversion networks for model-based optimization. In Advances in Neural Information Processing Systems 33, pp. 5126–5137, Virtual, 2020.

[3] Diffusion models for black box optimization. In Proceedings of the 40th International Conference on Machine Learning, pp. 17842–17857, Honolulu, HI, 2023.

**Questions:**

Please see Weaknesses part.

Besides, “They struggle with steering clear of out-of-distribution and invalid inputs” is often discussed in offline BBO instead of in online BBO. The reason why this paper presents it here for online setting should be discussed.

---

> ### Author Response · Authors · 2024-11-21
> **Response to Reviewer 1x7Q (1/3)**
>
> We thank the reviewer for the constructive comments and thoughtful suggestions. We address the concerns as follows:
>
> > The core difference between the main idea of the proposed method (the inverse method) and LLAMBO [1] should be explicitly discussed. While there are differences in the specific implementation details compared to LLAMBO.
>
> Thanks for sharing the LLAMBO paper. We have included it in the related work of our revised manuscript (line 88-90). Diff-BBO and LLAMBO are fundamentally different approaches in terms of **surrogate model and acquisition function design**, which can be well illustrated in Figure 1.
> * LLAMBO utilizes LLMs to improve Bayesian optimization. It belongs to forward surrogate modeling to estimate the likelihood $p(y|x,D)$ or $p(y \leq \tau|x,D)$, whereas Diff-BBO belongs to inverse surrogate modeling. This involves directly modeling the posterior $p(x|y,D)$ using the conditional diffusion model.
> * LLAMBO is proposed to deal with hyperparameter tuning problems where there are no validity concerns in the design space, meaning that every combination of hyperparameters in the hyperparameter space is valid. Diff-BBO is proposed to deal with the science and engineering tasks where the design space is high-dimensional and valid designs constitute a small subset of this space.
> * LLAMBO's acquisition function is a function of $x$.  The goal is to find the optimal $x$ in the design space. Conversely, the acquisition function in Diff-BBO is a function of $y$. The goal is to find the optimal objective $y$ for conditional sampling.
>
> >  this approach appears to be more of an aggregation of the previous methods. Specifically, the inverse approach was originally utilized in MINS [2], while the conditional diffusion model was incorporated in DDOM [3].
>
> Our paper presents three main contributions that distinguish Diff-BBO from LLAMBO, MINS, and DDOM:
> * We design an acquisition function Uncertainty-aware Exploration (UaE) for inverse modeling. **Theoretically, we prove that Diff-BBO using UaE achieves a near-optimal solution in online black-box optimization.**
> * We propose Diff-BBO, an inverse modeling approach for online black-box optimization that utilizes a conditional diffusion model, providing both **empirical significance** and **theoretical guarantees**.
> * We provide an uncertainty decomposition into epistemic uncertainty and aleatoric uncertainty for conditional diffusion model. We rigorously analyze how uncertainty propagates throughout the denoising process of conditional diffusion model.
>
> We would like to emphasize the takeaways of our theorems to form a better understanding for practitioners:
> * Quantifying sub-optimality: Theorem 2 establishes that the expected sub-optimality gap is bounded. This means that Diff-BBO provides a principled way to ensure that the generated candidates are close to optimal in expectation.
> * Controlling variance through uncertainty: Theorem 3 highlights that the variance of the sub-optimality gap is influenced by both aleatoric (inherent noise) and epistemic (model uncertainty) components. This decomposition underscores the importance of reducing epistemic uncertainty to stabilize the optimization outcomes and enhance reliability.
> * Balancing exploration and exploitation: Theorem 4 provides a theoretical foundation for the design of our acquisition function, Uncertainty-aware Exploration (UaE), which balances targeting high objective values and minimizing epistemic uncertainty. This balance enables practitioners to efficiently explore the design space while ensuring reliable convergence to optimal solutions.

---

> ### Author Response · Authors · 2024-11-21
> **Response to Reviewer 1x7Q (2/3)**
>
> > Superconductor, Ant and D’Kitty and so on are high dimensional problem. So, it would be better if this paper could compare with more high dimensional Bayesian optimization in recent years rather than simple black-box optimization.
>
> We have already compared Diff-BBO against multiple state-of-the-art black-box optimization baselines. **For high-dimensional Bayesian optimization baselines**, we included Trust Region BO (TuRBO)  [1] and Local Latent Space Bayesian optimization (LOL-BO) [2]. For acquisition function design, we included Joint Entropy Search (JEE) [3]. We also included Likelihood-free BO (LFBO) [4] and Conditioning by adaptive sampling (CbAS) [5] approaches. The details are provided in Section 6.2.
>
> > the paper does not sufficiently explore prior work in offline BBO. Expanding the discussion to include them could provide a more comprehensive literature review and motivation support.
>
> The problem setting of our paper is online BBO. **The research focus significantly diverges between online BBO and offline BBO.** Online BBO aims to **improve the sample efficiency of the sequential decision making algorithm**. In contrast, offline BBO assumes access to a fixed pre-collected dataset (line 53-55) and focuses on the surrogate modeling design.
>
> We mentioned the offline BBO setting in our paper primarily to contextualize the inverse surrogate modeling approach using deep generative models, which was originally introduced in the offline setting. However, the offline BBO setting itself is not directly relevant to our approach. Additionally, we have already discussed both inverse surrogate modeling and offline BBO [6][7][8][9][10][11][12][13] in the introduction and related work. We are happy to include additional relevant offline BBO references should there be any omissions.
>
> [1] Eriksson, David, et al. "Scalable global optimization via local Bayesian optimization." Advances in neural information processing systems 32 (2019).
>
> [2] Maus, Natalie, et al. "Local latent space bayesian optimization over structured inputs." Advances in neural information processing systems 35 (2022): 34505-34518.
>
> [3] Hvarfner, Carl, Frank Hutter, and Luigi Nardi. "Joint entropy search for maximally-informed Bayesian optimization." Advances in Neural Information Processing Systems 35 (2022): 11494-11506.
>
> [4] Song, Jiaming, et al. "A general recipe for likelihood-free Bayesian optimization." International Conference on Machine Learning. PMLR, 2022.
>
> [5] Brookes, David, Hahnbeom Park, and Jennifer Listgarten. "Conditioning by adaptive sampling for robust design." International conference on machine learning. PMLR, 2019.
>
> [6] Fu, Justin, and Sergey Levine. "Offline model-based optimization via normalized maximum likelihood estimation." arXiv preprint arXiv:2102.07970 (2021).
>
> [7] Kong, Lingkai, et al. "Diffusion models as constrained samplers for optimization with unknown constraints." arXiv preprint arXiv:2402.18012 (2024).
>
> [8] Krishnamoorthy, Siddarth, Satvik Mehul Mashkaria, and Aditya Grover. "Diffusion models for black-box optimization." International Conference on Machine Learning. PMLR, 2023.
>
> [9] Kumar, Aviral, and Sergey Levine. "Model inversion networks for model-based optimization." Advances in neural information processing systems 33 (2020): 5126-5137.
>
> [10] Li, Zihao, et al. "Diffusion model for data-driven black-box optimization." arXiv preprint arXiv:2403.13219 (2024).
>
> [11] Lu, Huakang, et al. "Degradation-Resistant Offline Optimization via Accumulative Risk Control." ECAI 2023. IOS Press, 2023. 1609-1616.
>
> [12] Trabucco, Brandon, et al. "Design-bench: Benchmarks for data-driven offline model-based optimization." International Conference on Machine Learning. PMLR, 2022.
>
> [13] Wang, Handing, et al. "Offline data-driven evolutionary optimization using selective surrogate ensembles." IEEE Transactions on Evolutionary Computation 23.2 (2018): 203-216.

---

> ### Author Response · Authors · 2024-11-21
> **Response to Reviewer 1x7Q (3/3)**
>
> > Why do the experimental tasks of online BBO methods use offline BBO benchmarks? Is it because there is no suitable benchmark for online BBO?
>
> There may be a misunderstanding. We did not use offline BBO benchmarks. Instead, we restructured 5 high-dimensional real-world tasks in biology, materials science, and robotics from Design-Bench to facilitate online black-box optimization. At each iteration, the online BBO algorithms query the oracle function to generate new data instead of using the Design-Bench offline dataset. We also included the new molecular discovery task. The details are provided in Section 6.1 and Appendix D.1. The reason we compared Diff-BBO with other baselines on these tasks is to verify the effectiveness of Diff-BBO on real-world science and engineering problems where the valid data represent a small subspace in the high-dimensional design space as we mentioned in Section 1 (line 41-44).
>
> > “They struggle with steering clear of out-of-distribution and invalid inputs” is often discussed in offline BBO instead of in online BBO. The reason why this paper presents it here for online setting should be discussed.
>
> The challenge of steering clear of out-of-distribution and invalid inputs does not depend on whether it is in an offline or online BBO setting. Indeed, it depends on the design space. When the design space is high-dimensional and valid designs constitute a small subset of this space, such optimization problems become exceptionally difficult, since the optimizer must avoid out-of-distribution and invalid inputs [9]. Consider an example of online BBO on this type of design space using Gaussian Processes. Once the designed acquisition function proposes a candidate in the design space, the forward surrogate model lacks the capability to determine whether this candidate is valid without explicit constraint information.
>
> This type of optimization problem is common in real-world science and engineering tasks. We conducted 6 such tasks and verified that Diff-BBO utilizing the conditional diffusion model for inverse modeling can capture data distributions in the design space, facilitate optimization within the data manifold, and therefore outperforms state-of-the-art forward approaches in terms of sample efficiency.
>
> [9] Kumar, Aviral, and Sergey Levine. "Model inversion networks for model-based optimization." Advances in neural information processing systems 33 (2020): 5126-5137.

---

> > ### Author Response · Authors · 2024-11-27
> > **The end of the rebuttal phase is approaching**
> >
> > Dear Reviewer 1x7Q,
> >
> > As the end of the rebuttal phase is approaching, we would like to kindly confirm if our responses have successfully addressed your concerns and clarified your questions. If you have any additional questions or concerns, we would be happy to address them.
> >
> > Best regards,
> >
> > Authors of Diff-BBO

---

> ### Comment · Reviewer_1x7Q · 2024-12-03
> **Thanks for the feedback**
>
> Thanks for the feedback and I have read the rebuttal.
>
> The rebuttal addresses some of my concerns. However, the concerns of experiments in the review have not been fully addressed.
>
> Besides, from the rebuttal, it involves many issues: online optimization, offline optimization, OOD, high-dimension optimization, and constrained optimization, which makes the motivation and presentation of this paper become more unclear. The evidence and experiments in the current version cannot fully support these issues and motivations.
>
> Best regards

---

> > ### Author Response · Authors · 2024-12-04
> >
> > Our paper maintains a clear focus on **online black-box optimization** (BBO). We believe the evidence and experiments presented in the paper sufficiently support our claims and align with the motivations outlined. Hence, we respectfully disagree with the reviewer’s response above. While we appreciate the reviewer’s feedback, we stand by the clarity and focus of our contributions and will continue to refine the paper to further emphasize its strengths in future iterations.

---

### Author Response · Authors · 2024-11-27
**General Response**

We thank all the reviewers for the insightful questions and feedbacks. We appreciate Reviewer DvmK, Reviewer cYdB, and Reviewer TAkd pointing out that our method is well-motivated, well-written, and supported by strong empirical results. We also thank Reviewer DvmK and Reviewer TAkd for highlighting the theoretical soundness of our approach.

We have carefully addressed each question, conducted additional experiment requested, and provided more detailed theoretical insights to form a better understanding for practitioners.

**Additional Experiment**:

* Ablation study on the factor weight $\beta$ of the epistemic uncertainty term in the acquisition function Uncertainty-aware Exploration (UaE), providing insights into the significance of uncertainty in the acquisition process.

**Theoretical Insights**:

* Quantifying sub-optimality: Theorem 2 establishes that the expected sub-optimality gap is bounded. This means that Diff-BBO provides a principled way to ensure that the generated candidates are close to optimal in expectation.
* Controlling variance through uncertainty: Theorem 3 highlights that the variance of the sub-optimality gap is influenced by both aleatoric (inherent noise) and epistemic (model uncertainty) components. This decomposition underscores the importance of reducing epistemic uncertainty to stabilize the optimization outcomes and enhance reliability.
* Balancing exploration and exploitation: Theorem 4 provides a theoretical foundation for the design of our acquisition function, Uncertainty-aware Exploration (UaE), which balances targeting high objective values and minimizing epistemic uncertainty. This balance enables practitioners to efficiently explore the design space while ensuring reliable convergence to optimal solutions.

We addressed the individual questions of each reviewer below in specific replies. Please do not hesitate to inform us if you have any additional concerns or suggestions. We sincerely appreciate your valuable insights and feedbacks throughout the review process. Thank you once again for your time and effort in reviewing our work.

---

### Meta-Review · Area_Chair_Zgi9 · 2025-01-01

**Metareview:**

A summary of the strengths and weaknesses based on the reviews and rebuttal (including the follow-up discussion and that among the reviewers) is provided below:

**STRENGTHS**
- The paper provides an interesting approach with theoretical performance guarantees and empirical validation.
- The paper is well-written and easy to follow.

**WEAKNESSES**
Some major concerns remain:

- How does the ad hoc way of constructing the candidate set Y in practice (mentioned in the rebuttal) affect the theoretical results and the sensitivity of the empirical performance (Reviewer DvmK)? On the other hand, the authors say in the proof of the theoretical results (line 1083) that the candidate set Y is constructed based on the model's predictions and is designed to explore the objective space efficiently. How can we reconcile what the authors say in the rebuttal with that said in the paper?

    The authors say in the rebuttal that "$\phi_k$ is a predetermined constant ($\phi_k$ is the maximum function value being queried in the current training dataset (line 241-242))". Since the current training dataset changes in each iteration, why does  remain a predetermined constant? If not, can the authors explicitly discuss how the theoretical results are being affected by the changing candidate set Y in each iteration?

- While this paper focuses on the problem setting of online BBO which is different from that of offline BBO (such as assuming access to a prior dataset), both settings utilize the inverse approach and the conditional diffusion model.

    It is not clear what then would be the additional nontrivial technical challenges from extending the solution framework from the offline setting to solve the online setting, considering that both settings utilize the inverse approach and the conditional diffusion model. This could be the main concern of Reviewer 1x7Q (specifically, Weaknesses 1 and 3).

    Furthermore, would it be possible to simply rerun an existing approach to offline BBO in each iteration, while considering the collected data from previous iterations to be the prior dataset? Considering that a relatively large prior dataset is also being made available to the online BBO (see rebuttal to Reviewer DvmK: in practice, is it reasonable to assume that it only include data with below-average objective scores?) and that a relatively large batch size is being considered in each iteration, the distinction between the problem settings between online and offline BBO seems to be blurred. We would also recommend that the authors compare the model quality across iterations in their experiments to be able to accurately say that the model benefits from the updates.

- Considering that the authors have obtained better performance with a large batch size, it would have been more informative to consider giving theoretical results that account for the batch size N as a parameter (suggestion of Reviewer cYdB) and whether doing so can yield interesting or practical insights. This does not imply that the current results in the paper are not sound.

- (Minor point) The authors should include the results with a factor of beta (Reviewers cYdB and TAkd) in the main paper, considering that beta = 1.2 instead of 1.0 gives better outcome. How does the inclusion of beta (not equal to 1) as a parameter affect the theoretical results?

The authors are strongly encouraged to revise their paper based on the above feedback and that of the reviewers.

**Additional Comments On Reviewer Discussion:**

See above.

---

### Decision · Program_Chairs · 2025-01-22

Reject